# Recent Advances in Optically Controlled PROTAC

**DOI:** 10.3390/bioengineering10121368

**Published:** 2023-11-28

**Authors:** Muzi Ouyang, Ying Feng, Hui Chen, Yanping Liu, Chunyan Tan, Ying Tan

**Affiliations:** 1State Key Laboratory of Chemical Oncogenomics, Institute of Biomedical and Health Engineering, Shenzhen International Graduate School, Tsinghua University, Shenzhen 518055, China; oymz23@mails.tsinghua.edu.cn (M.O.); fengy22@mails.tsinghua.edu.cn (Y.F.); hui-chen21@mails.tsinghua.edu.cn (H.C.); yanpingliu@sz.tsinghua.edu.cn (Y.L.); tancy@sz.tsinghua.edu.cn (C.T.); 2Department of Chemistry, Tsinghua University, Beijing 100084, China

**Keywords:** PROTAC, targeted degradation, optogenetics, NIR

## Abstract

Proteolysis-targeting chimera (PROTAC) technology is a groundbreaking therapeutic approach with significant clinical potential for degrading disease-inducing proteins within targeted cells. However, challenges related to insufficient target selectivity raise concerns about PROTAC toxicity toward normal cells. To address this issue, researchers are modifying PROTACs using various approaches to enhance their target specificity. This review highlights innovative optically controlled PROTACs as anti-cancer therapies currently used in clinical practice and explores the challenges associated with their efficacy and safety. The development of optically controlled PROTACs holds the potential to significantly expand the clinical applicability of PROTAC-based technology within the realm of drug discovery.

## 1. Overview of Cellular Targeted Protein Degradation (TPD) Processes

Chemotherapeutic treatments have long been a cornerstone of cancer therapy. Nevertheless, contemporary conventional small molecule inhibitor-based chemotherapeutic drugs possess two obvious drawbacks affecting their selectivity and effectiveness. First, they are frequently unable to efficiently bind to cellular targets with low concentrations, limiting their abilities to directly regulate the function of proteins of interest (POIs) [1,2]. Second, high doses of these drugs can lead to drug resistance and other side effects [3,4,5]. In response to these challenges, researchers have explored ways to harness the power of cellular degradation systems to remove disease-inducing target proteins from cells as an innovative anti-cancer therapeutic strategy.

Eukaryotic cells employ two protein degradation pathways: the ubiquitin-proteasome system (UPS)-dependent pathway (Figure 1) and the lysosome-mediated proteolytic pathway. The UPS-dependent pathway serves as the core cellular system for degrading over 80% of intracellular misfolded or damaged proteins via a two-stage process [6]. In the first stage [7], a high-energy thiol-ester bond forms between a ubiquitin-activating enzyme (E1) and ubiquitin (Ub) to activate Ub. Activated Ub is then transferred to the ubiquitin-conjugating enzyme (E2), which, under the guidance of ubiquitin ligase (E3), transfers Ub to the target protein. E3 catalyzes the formation of a covalent isopeptide bond, linking the Ub molecule to a lysine residue on the target protein. Subsequently, E3 attaches an additional Ub molecule to the first one, repeating this ATP-dependent process to achieve the polyubiquitination of the target protein. In the second stage, the polyubiquitinated protein undergoes hydrolysis by the 26S proteolytic enzyme complex, generating oligopeptides that are released from the proteasome. During this process, the action of the deubiquitinating enzymes induces Ub dissociation from the substrate and subsequent return of Ub to the cytoplasm for reuse.

Humans possess two types of E1, approximately 40 types of E2, and more than 600 types of E3 ligases. A variety of TPD methodologies based on UPS have been developed by researchers, with PROTAC and molecular glue among the most well-established. In addition to the UPS-dependent pathway, eukaryotic cells employ an alternative protein degradation pathway known as the autophagy-lysosomal pathway. This pathway is responsible for the degradation of a diverse array of substrates, including aging or damaged organelles, protein aggregates, and invasive pathogenic microorganisms [8,9]. Recently, researchers have explored harnessing this pathway to achieve the targeted protein degradation (TPD) of POIs. These efforts have led to the development of methodologies incorporating autophagosome-tethering compounds (ATTECs) [10], autophagy-targeting chimeras (AUTACs) [11], auto TAC [12], and other innovative approaches with potential clinical value.

Recent advances in TPD technology have significantly expanded the range of viable targets, while also providing powerful methodologies to address therapeutic challenges that cannot be addressed using conventional small molecule inhibitors of the protein function. This review primarily focuses on the mechanistic aspects of PROTACs for achieving TPD, provides an overview of the current TPD research landscape, and examines the clinical feasibility of optically controlled PROTACs for the targeted removal of disease-inducing proteins from cells to alleviate various pathological disorders.

## 2. Overview of PROTACs

### 2.1. Structure and Advantages of PROTACs

The concept of PROTACs was first proposed by Crews et al. in 2001. A PROTAC molecule consists of a protein of interest (POI) ligand, an E3 ligand, and a linker (Figure 2). PROTAC systems mainly target cytoplasmic POIs that can bind to PROTAC molecules through covalent, non-covalent, or allosteric interactions. Consequently, POI ligands must be designed to specifically bind to POI structural features in order to achieve the effective targeting of POIs. Although more than 600 human genome-encoded E3 ubiquitin ligases have been identified to date, only four of them have been integrated into the most advanced PROTAC systems: mouse double minute 2 (MDM2), cereblon (CRBN), Von Hippel–Lindau (VHL), and cLAP E3 ligases [13]. Importantly, linker composition, structural characteristics, and length have been reported to influence PROTAC system functionality and stability. In general, PROTAC linkers consist of 10 to 20 atoms, with polyethylene glycol (PEG) or alkyl chains utilized as connecting structures that endow PROTACs with enhanced flexibility [14]. Additionally, rigid chains containing piperidine and piperazine residues can be incorporated within linkers to increase PROTAC stability [15].

Traditionally, the effectiveness of most small molecule drugs relies on their ability to bind to the active site of enzymes or receptors, a mechanism known as occupation-driven mode. However, PROTAC offers a unique approach by degrading proteins through any accessible region or crevice, without the requirement of occupying an active pocket. This event-driven pharmacological mechanism distinguishes PROTAC from conventional drugs, as it does not rely on target occupancy to disrupt the function of the target protein. Consequently, PROTAC shows great potential in targeting proteins with inactive sites, expanding the possibilities for the development of previously untargetable protein targets. Moreover, PROTACs can achieve therapeutic effects by degrading entire proteins [16,17,18,19,20], rather than individual protein domains that are typically targeted by small molecule inhibitors. These advantageous PROTAC characteristics have prompted medical researchers to explore their chemotherapeutic potential, resulting in the identification of many PROTAC targets with potential clinical applicability to human disease treatment.

### 2.2. Current Disadvantages of PROTACs

Nonetheless, while PROTAC technology has invigorated the field of drug development, PROTACs come with certain limitations that warrant attention. First, their exceptionally high molecular weights and poor cell permeability can lead to patent-related issues. Second, unique PROTAC ternary compound structures can induce a hook effect (the phenomenon of false negatives due to the inappropriate ratio of antigens to antibodies) which, at high concentrations, can lead to the formation of non-functional binary compounds. Third, the expression levels of E3 ligase vary across different cells and tissues, making it challenging to optimize E3 ligase effects. Fourth, PROTAC targeting lacks the necessary specificity to prevent PROTAC-induced TPD from occurring in normal cells and tissues that could trigger adverse side effects.

In recent years, significant efforts aimed at enhancing the targeting specificity of PROTACs have resulted in the development of numerous innovative PROTAC variants, including antibody-based targeting chimeras (AbTACs) [21,22,23], optically controlled PROTACs, folate-caged PROTACs [24], APtamer-PROTAC conjugates (APCs) [25], and others. These novel PROTACs not only offer improved targeting specificity but also exhibit greater stability and unique on-off capabilities. These attributes empower them to exert temporal and spatial control over the degradation of POIs, recruit new E3 ligases, and achieve their intended effects with reduced toxicity to normal cells. These advancements are discussed in greater detail below.

### 2.3. The Evolution of PROTACs

The primary PROTACs utilized degradation-derived peptides, such as phosphopeptides [26] and hydroxyl peptides [27,28,29] to recruit endogenous E3 ubiquitin ligases. Peptide-based PROTAC has the advantages of being easy to modify and having a larger contact area than small-molecule drugs, but it has high molecular weight and low delivery efficiency and cannot really function as a drug. It can only enter the target cells through injection therapy, and its clinical value is not high. In 2018, Jiang et al. [30] optimized peptide-based PROTAC and designed a stable peptide-based PROTAC. The heterobifunctional peptide (TD-PROTAC) was formed by teaming an N-terminal aspartic cross-linked stable peptide ERα modulator (TD-PERM) with pentapeptides binding a VHL E3 ubiquitin ligase complex, which had better stability and cell permeability, and was able to degrade ERα in vitro and in vivo.

The second generation of PROTAC is based on some specific E3 ligands, such as TIR E3 ligase auxin [31] and nutlin for mouse double minute 2 homolog (MDM2) E3 ligase nutlin [32], to develop small molecule PROTACs. For example, the first nutlin-based small molecule PROTAC developed by Schneekloth et al. [33] in 2008 was used to target the degradation of the androgen receptor AR.

Due to the disadvantages of the first two generations of the PROTACs mentioned above, the development of the next generation of PROTACs urgently needs to improve the targeting specificity to mitigate the problem of cytotoxicity.

## 3. Optically Controlled PROTACs

To address the issue of PROTAC off-target toxicity, a “prodrug design” strategy has been proposed that generates specialized PROTACs (pro-PROTACs) that can be selectively activated in target tissues. Pro-PROTACs are engineered by introducing stimulus-responsive cage motifs into parent PROTAC molecules at crucial binding sites within heterobifunctional molecules to hinder the PROTAC ubiquitination of POIs. For example, numerous studies have underscored the benefits of attaching glutathione-cleavable aptamers [34], folic acid [35], or enzyme-responsive motifs [36] to the hydroxyl group of the E3 ligand to control the PROTAC activation state. These endogenous modifications act as “on-off” switches that control PROTAC functionality through the regulation of PROTAC ternary complex formation. However, the activation of this control mechanism can be influenced by the tumor microenvironment, making it difficult to engage the switch with pinpoint accuracy. In contrast, exogenous stimuli offer a more precise and readily controllable means of “flipping” the switch to achieve precise on-demand PROTAC activation.

The development of methodologies for harnessing light as a non-invasive external stimulus has resulted in the emergence of photodynamic therapy (PDT), a powerful approach for achieving precise drug delivery to target cells and tissues. PDT systems employ light as an external controller to activate PROTACs with temporospatial precision, thereby enhancing the specificity of the PROTAC-mediated degradation of POIs.

Light-controlled PROTACs primarily fall into two categories: photocaged PROTACs (pc-PROTACs) and photoswitchable PROTACs (photoPROTACs). These variants are typically created by incorporating light-sensitive molecular groups into the parent PROTAC structure. In the absence of light, steric hindrance between introduced photosensitive components and parent PROTAC structures prevents interactions from occurring between the E3 ligase and the POI or between the POI and the PROTAC, thereby maintaining the PROTAC in an inactive state. However, exposure to light activates these photosensitive components, causing them to swiftly dissociate from PROTAC molecules. Consequently, the PROTAC’s ability to assume its active conformation is restored, enabling the PROTAC to induce the degradation of POIs.

### 3.1. Photocaged PROTACs

Photocleavable protecting groups (PPG) are groups of photosensitive compounds that can be selectively removed from molecules through irreversible photocracking reactions under specific light conditions. In the context of PROTACs, photocaged PROTACs involve incorporating light-responsive protective groups into the E3 ligand or target protein ligand. Consequently, the targeted degradation activity of the PROTAC molecule is effectively blocked. By applying light, the protective group is cleaved, thereby the target protein is degraded.

#### 3.1.1. Photocaged PROTACs Based on CRBN

Further investigation of the crystal structure of CRBN and o-phenylenediamine complexes revealed that the pentenediamine nhh in o-phenylenediamine is essential for its binding to CRBN, and the caging of glutarimide NH with methyl groups or other groups eliminates pomalidomide’s ability to bind to CRBN E3 ligase. At present, several CRBN-based pc-PRPTACs have been reported, including opto-PROTACs, pc-PROTACs, and so on.

In 2019, Pan’s research group [37] created a novel PROTAC by introducing the bulky 4,5-dimethoxy-2-nitrobenzyl (DMNB) photocage group into a PROTAC molecule to generate the first photocaged PROTAC. They firstly developed a photocaged PROTAC molecule 1 (pc-PROTAC1) (Figure 3B) by introducing a photo-controlled group into a PROTAC molecule (dBET1) capable of degrading the bromodomain-containing protein 4 (BRD4). After light exposure, pc-PROTAC1 produced dBET1, which effectively degraded the BRD4 protein in cells and zebrafish. Thus, this work established a general strategy for using light-induced protein degradation.

The introduced DMNB photocage group disrupted the PROTAC structure via a steric hindrance-based mechanism, preventing the PROTAC from inducing degradation on the target POI, bromodomain-containing protein 4 (BRD4). Importantly, the triggering of POI degradation could be controlled by exposing the PROTAC to ultraviolet light of a specific wavelength (365 nm) to induce the release of the DMNB photocage from the PROTAC, thereby restoring PROTAC’s ability to bind to its target protein. The success of this approach prompted other research teams to explore the regulatory role of the DMNB photocage motif DMNB in controlling PROTAC functionality. Consequently, this photocage motif has been incorporated in numerous photocaged PROTACs for the targeting of various POIs (Figure 3A).

Using a different approach, Liu et al. successfully created a novel type of adaptable photocaged PROTAC capable of precisely triggering the degradation of POIs both temporally and spatially. Their opto-PROTAC system was created by covalently coupling the parent PROTAC to the glutarimide amino group of the photocage molecule pomalidomide. Consequently, the modified PROTAC was unable to recruit the CRBN E3 ligase to effect TPD until it was exposed to light, which triggered the release of pomalidomide to restore PROTAC function. Due to the successful use of pomalidomide as an E3 ligand in various recently developed photocaged PROTACs, researchers have explored the use of pomalidomide-derived PROTACs to achieve the controlled degradation of other POIs, such as cyclin-dependent kinase [38,39], the BCR-ABL [40] fusion protein, Tau [41], and others.

Notably, this photocaged PROTAC system can be customized to target various E3 ligases by incorporating ligands other than pomalidomide. For example, Naro et al. [42] employed a similar approach to develop caged PROTACs capable of achieving the optical activation of small molecule-induced TPD. Their opto-PROTAC was able to recruit two E3 ligases, VHL and CRBN, due to the incorporation of two distinct light-controllable protective groups (cage groups) containing two different E3-binding ligands. This method is suitable for targeting POIs that can be bound by these E3 ligases, thus expanding the potential clinical applicability of opto-PROTACs as chemotherapeutic treatments for a wide range of diseases.

#### 3.1.2. Photocaged PROTACs Based on VHL

In addition to CRBN-based PROTACs, VHL-based PROTACs are another major class of photocaged PROTACs.

Kounde et al. [43] developed a conditional PROTAC using caged VHL ligands and light as a conditional stimulus. They attached the DMNB group to the VHL E3 ligand, activated PROTACs for a short period of time (60 s), and then targeted the degradation of the BRD4 protein, where E3 ligase activity depends on light.

In another study by Naro et al. [42], VHL-based PROTACs were caged by photoclavated diethylamine maroumarin (DEACM) to degrade ERRα. The resulting caged PROTACs were initially inert and regained their ability to target degradation after UVA (λ = 360 nm) activation.

The addition of photocage groups only affects the binding of PROTACs to VHL E3 ligase, but not to the target protein substrate. Therefore, these methods of photocage mentioned above can also be applied to other VHL-based PROTACs.

The X-ray is an important radiation therapy strategy for cancer, with excellent precision and deep tissue penetration. Local X-ray therapy can induce tumor necrosis through DNA damage, apoptosis, and cell necrosis. In order to control the potential systemic toxicity of PROTACs caused by protein degradation in unexpected tissues, Yang et al. [44] reported an X-ray radiation-triggered PROTAC prodrug (RT-PROTAC), which combines the azide phenyl-cage groups into the design of PROTAC molecules to form X-ray responsive RT-PROTACs. The target protein degradation is precisely controlled spatially by X-rays. RT-PROTACs remain inert before X-ray radiation and are activated by X-ray radiation to target protein degradation in vivo.

#### 3.1.3. Other Kinds of Optically Controlled PROTACs

Ko et al. [45] developed a spatiotemporal precision photochemically targeted chimera (PHOTACs), which catalyzes the ubiquitination and degradation of target proteins at specific wavelengths of light. They designed PHOTACs to target the degradation of Ca^2+^/calmodulin-dependent protein kinase IIα (CAMKIα), a protein kinase essential for the synaptic function of excitatory neurons. The authors demonstrated in mouse brain tissue that the light activation of the Camkiα-photac in a region 25 μm from the brain surface measured synaptic function by light-induced attenuation of the evoked field excitatory postsynaptic potential (fEPSP) response to physiological stimuli and found that the light-controlled degradation of the Camkiα-photac reduced synaptic function within a few minutes. This PHOTACs method is widely applied to other key proteins involved in synaptic function, in particular, to assess their role in long-term enhancement within subcellular dendrite domains and in memory maintenance.

Zhang et al. [46] reported a strategy to block checkpoint signaling pathways through the combination of nano-PROTACs with photodynamics for the photoimmunotherapy of cancers. These nano-PROTACs consist of photosensitizers (protoporphyrin IX, PpIX) and phosphatase 2 (SHP2) containing the Src homologous 2 domain targeting PROTAC peptides (aPRO) via cleavable fragments of caspase 3. After irradiation, the expression of caspase 3 in tumor cells was increased and activated, and the ubiquitination degradation of SHP2 was induced. The persistent loss of SHP2 blocks the immunosuppressive checkpoint signaling pathway (CD47/SIRRP α and PD-1/PD-L1), thereby reactivating anti-tumor macrophages and T cells. This study may represent a universal ROTAC platform for P to modulate immune-related signaling pathways.

#### 3.1.4. Near-Infrared Light Triggered Photocaged PROTAC System

Although photocaged PROTACs have been widely used for various applications, they possess poor stability due to the frequent detachment of the photocage group from the parent PROTAC molecule. Currently, most photocaged PROTACs are activated by ultraviolet light (365 nm), which has limited tissue-penetration ability and can only be applied effectively to a few cancer types, such as blood and skin cancers. Additionally, ultraviolet exposure can induce DNA damage in living organisms [47,48], an undesirable phototoxic effect. To overcome these limitations, researchers have been exploring cage groups that can absorb other wavelengths of light, particularly those within the near-infrared (NIR) spectrum.

Building on this idea, He et al. designed a photocaged PROTAC known as phoBET1. PhoBET1 incorporates lanthanide-doped upconversion nanoparticles (UCNPs) capable of converting NIR light of excitation wavelength 980 nm into emitted ultraviolet light. The packaging of phoBET1 molecules in silicon dioxide nanoparticles (UMSNs) resulted in the creation of NIR light-activatable PROTAC nanocages (UMSNs@phoBET1) [49]. Experimental assessments of the UMSNs@phoBET1 protein degradative and apoptotic effects in vivo demonstrated that the exposure of these nanocages to NIR light at the above-mentioned wavelength could achieve the targeted degradation of BRD4 to induce time-dependent apoptosis of MV4-11 cancer cells.

Using a different approach, Zhang’s research group developed a nano-PROTAC that was activatable by NIR light (NAP) [50]. NAP was created through the self-assembly of complex molecules composed of amphiphilic PROTACs coupled to a cleavable linker consisting of a singlet oxygen (^1^O_2_) atom-containing NIR photosensitizer. Upon systemic administration, the non-activated NAP selectively accumulated within the tumor tissues. Subsequent irradiation of tumors with NOR light triggered the release of the NIR photosensitizer from the PROTAC molecules to thereby restore PROTAC activity that enabled the PROTAC to recruit E3 ligase and perform its BRD4-degrading function.

The NIR light-activatable PROTAC nanoplatform addresses the limitations of existing shortwave photo-controlled PROTACs by offering the precise regulation of PROTAC activity within living tissues. This technology holds great potential to revolutionize chemotherapeutic treatments by providing tools for achieving improved tumor targeting without harming healthy tissues.

### 3.2. Photoswitchable PROTACs

Unlike conventional PROTACs, photoswitchable PROTACs (photoPROTACs) employ ligands containing azo groups instead of alkyl or polyether groups to link POIs to E3 ligases. When exposed to specific wavelengths of light, these photoPROTACs undergo reversible photoisomerization between “cis” and “trans” isomeric forms, enabling precise and bidirectional control over PROTAC molecular degradation activity (Figure 4A). This regulatory effect depends on the predictable effects of PROTAC molecular conformation changes on protein degradation activity.

The innovative concept of achieving continuous spatiotemporal control over the degradation of POIs was first introduced by Pfaff et al. [51] as a strategy to minimize the systemic toxicity associated with chemotherapy. They designed photoPROTACs that contained a linker composed of ortho-F4-azobenzene ligands inserted between two warhead ligands (Figure 4B). Consequently, PROTACs containing this linker were able to undergo photoisomerization when exposed to light at wavelengths of 530 nm and 415 nm, which induced the generation of a long-lasting state of photohomeostasis (PSS) in the absence of thermally induced reverse isomerization. PhotoPROTAC-based strategies offer advantages over traditional irreversible activator release-based photocage strategies and, thus, are attracting increasing research interest.

Reynders et al. introduced the concept of photopharmacology to the field of PROTAC [52] technology by designing an optically controllable PROTAC variant (PHOTAC). PHOTACs incorporate azobenzene optical switches that allow the controlled regulation of their activation to be achieved through exposure to specific wavelengths of blue-violet light (380 to 440 nm). Once activated, PHOTACs effectively induce degradation on various targets, including FK506-binding protein 12 (FKBP12) and BRD2-4 proteins, by binding to the CRL4^CRBN^ E3 ligase complex to effect the proteolysis of POIs in a light-dependent manner.

Jin et al. developed photoswitchable Azo-PROTACs for azobenzene proteolytic targeting by incorporating azophenyl groups between the ligand of the E3 ligase and the target protein [53]. Azo-PROTACs are small molecule tools that can be controlled by light to selectively eliminate proteins in cellular systems. In a study utilizing the lenalidomide-azo-Dasatinib three-functional system, it was demonstrated that the trans-isomer and cis-isomer of azo-PROTAC exhibited distinct protein degradation activities. The researchers successfully manipulated the degradation of BCR-ABL proteins by modulating the configuration of Azo-PROTAC through UVC light. Importantly, they also verified that the active state of Azo-PROTAC could be switched on and off in living cells via ultraviolet irradiation, providing further evidence of its controllable behavior.

As an additional innovation, Zhang et al. [54] achieved the spatiotemporal control of the PROTAC-induced degradation of POIs through the development of novel PROTACs known as arylzolpyrazole optical switch PROTACs (AP-PROTACs). These AP-PROTACS were the first photoPROTACs that could target the degradation of multiple protein kinases through the incorporation of a hybrid kinase inhibitor ligand capable of binding to four different protein kinases. Importantly, the on/off activation of this AP-PROTAC could be controlled through the exposure of this PROTAC to two different wavelengths of light.

## 4. Current Defects of Optically Controlled PROTACs

In addition to the above-mentioned activation or inactivation of optically controlled PROTACs with UVA, UVA may induce oxidative stress and DNA damage, as well as limited tissue penetration of UVA (photocaged PROTAC and photoPROTACs) and visible light (photoPROTACs). Light-controlled PROTAC is only suitable for a few cancers that are easily exposed to light, such as skin cancer or leukemia. Therefore, further research should focus on using other light sources, such as near-infrared light instead of ultraviolet light, to trigger optically controlled PROTACs.

Another potential disadvantage of optically controlled PROTACs is their poor permeability, as well as drug delivery. In contrast to small molecule drugs, which are usually less than 500 Da [55], the average PROTAC is usually above 600 Da. The molecular weight of optically controlled PROTACs is higher, usually around 1000 Da. A larger molecular weight will affect the pharmacokinetics and delivery efficiency of optically controlled PROTACs. Therefore, there is an urgent need to optimize the drug performance of PROTACs, so that it can be administered orally or develop other delivery methods with convenient and efficient delivery.

We know that, as event-driven molecules, PROTACs are not consumed throughout the protein degradation process, but act as a catalyst-like component. Therefore, how to accurately stop degradation is a problem that needs to be solved. Recently, researchers proposed a method of “ligation to scavenging” [56] to achieve the controlled termination of the PROTAC protein degradation process. The termination system consists of PAMAM-G5-TCO and tetrazine-modified PROTACs (Tz-PROTACs). Among them, PAMAM-G5-TCO is modified by functional trans-cycloctene (TCO) molecules with the dendritic macromolecule PAMAM as the carrier. The modified PAMAM-G5-TCO has a spherical structure, is rigid, has good biocompatibility, and can rapidly clear the free tetraazine fragment labeled PROTACs molecules in the cell through the reverse electron demand Diels–Alder reaction (IEDDA), thereby stopping the degradation of the targeted proteins in the cell. This study enables a chemical knockdown method to precisely adjust the level of POI in living cells, which may be combined with light control to provide a new way to accurately control the degradation of target proteins in space and time.

In this current study, the molecules of light-photocaged PROTACs and photoswitchable PROTACs are based on the development of two ligases: VHL and CRBN. In the future, new light-controlled PROTACs can be developed on the basis of the two E3 ligases, MDM2 and cLAP, which may produce different opportunities.

## 5. Summary and Prospect

PROTACs operate in an “event-driven” mode by leveraging the body’s inherent protein disposal system, the UPS, to fight disease by reducing expression levels of disease-causing target proteins rather than merely inhibiting their functions. While several PROTAC molecules have been shown to exert beneficial anti-tumor effects in clinical trials, concerns persist regarding their lack of targeting specificity and toxicity toward healthy tissues.

In recent years, researchers have made significant progress in developing optically controlled PROTAC components, such as photocaged PROTACs and photoPROTACs that may potentially reduce chemotherapy-associated toxic side effects and enhance drug targeting precision. These innovative approaches utilize light as a specialized external control agent, allowing for the light-mediated temporospatial activation of PROTACs to improve their target selectivity.

The successful development of optical switches and light cage PROTACs has enabled the precise control and fine-tuning of PROTAC activity. Notably, optical switch PROTACs offer the significant advantage of reversibility, allowing them to transition between active and inactive states, whereas photocaged PROTACs can only convert from inactive to active structures. Furthermore, the use of light as an activation signal offers the benefits of low toxicity and high spatial and temporal resolution that should enable the rapid advancement of the PROTAC field and the realization of PROTACs’ clinical therapeutic potential. Future research endeavors should focus on developing PROTACs with light excitation wavelengths within the visible and near-infrared spectral regions to enhance PROTAC tissue-penetrating capabilities and safety.

And, there are still many questions about how optically controlled PROTACs penetrate cell membranes and have clinical medicinal value and further studies are needed to understand their uptake, metabolism, excretion, and toxicity. Maintaining the required concentration of PROTACs’ pharmacological effects and improving cell absorption and bioavailability is a key issue that needs to be solved. In order to overcome these challenges, it is important to select linking molecules with the best physical and chemical properties. The target protein FKBP12 is susceptible to protein degradation caused by the chemical RC32 [57], highlighting the potential to improve PROTACs with suitable linkers to increase their permeability. In the future, we can improve the structure of the linker to increase the transmission efficiency of optically controlled PROTACs.

In summary, the development of chemotherapeutic treatments incorporating optically controlled PROTACs holds promise for delivering enhanced anti-tumor therapies with improved tumor specificity and reduced toxicity compared to the corresponding features of conventional therapies. These innovations pave the way for safer and more effective treatments for cancer and tumor-related diseases.

## Figures and Tables

**Figure 1 bioengineering-10-01368-f001:**
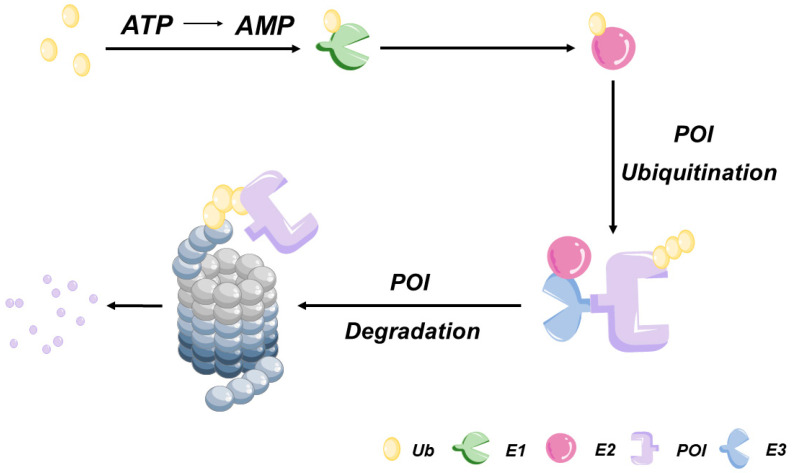
Ubiquitin-proteasome system (UPS). Firstly, the ubiquitin activating enzyme (E1) forms a thioester bond with ubiquitin in a reaction that requires ATP. Secondly, ubiquitin is transferred to ubiquitin transferase (E2) to form a new high-energy thioester bond. Finally, ubiquitin ligase (E3) provides substrate recognition and catalyzes covalent binding of ubiquitin to the target substrate through isopeptide bonding. The polyubiquitinated substrate is transported to the 26S proteasome for degradation.

**Figure 2 bioengineering-10-01368-f002:**
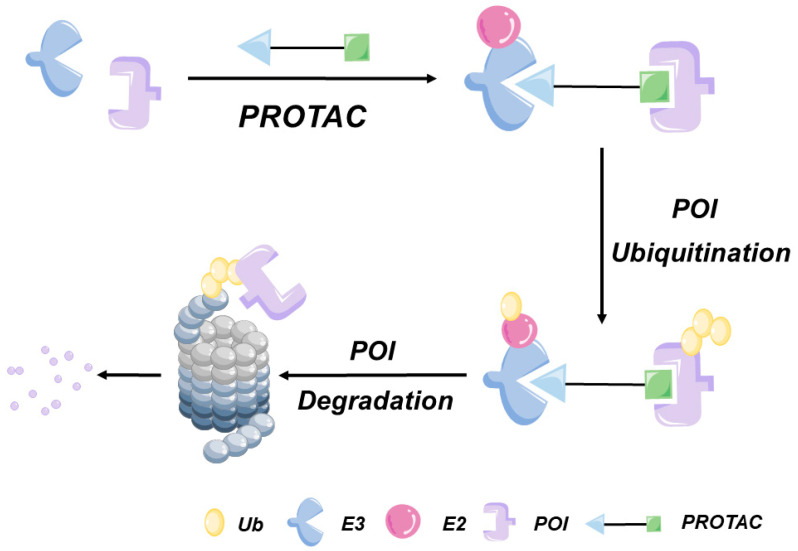
Proteolytic targeted chimera (PROTAC). PROTAC is a heterobifunctional compound composed of target protein ligand and E3 ligand. The simultaneous binding of target protein and E3 ligase promotes the formation of ternary complexes. PROTAC promotes the degradation of target protein through UPS pathway.

**Figure 3 bioengineering-10-01368-f003:**
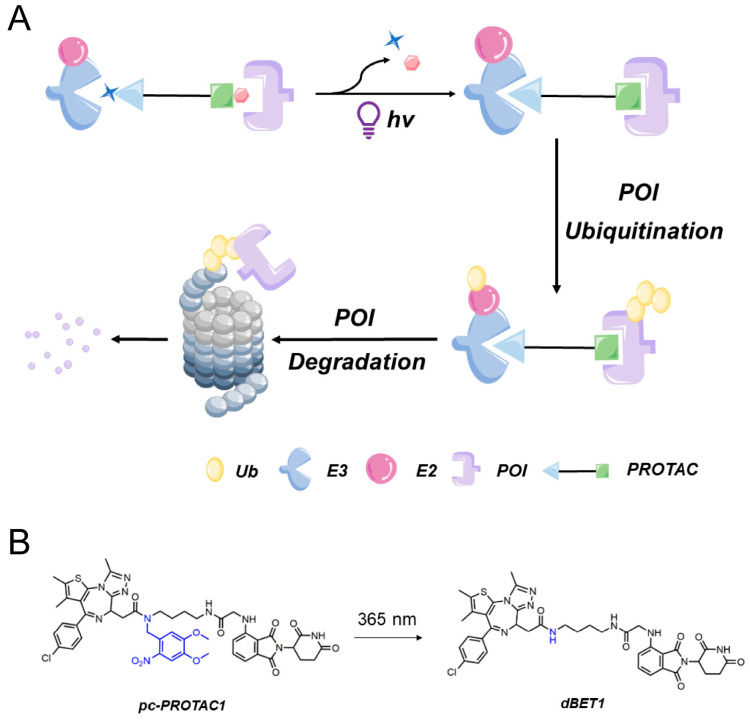
(**A**) Photocaged PROTACs. Photocaged PROTACs incorporate photoresponsive protective groups into the E3 ligand or target protein ligand, and PROTAC is inert in the absence of light. After exposure to light, the photocage is shed and activated PROTAC degraded the target protein. (**B**) Deinclusion reaction of pc-PROTAC1.

**Figure 4 bioengineering-10-01368-f004:**
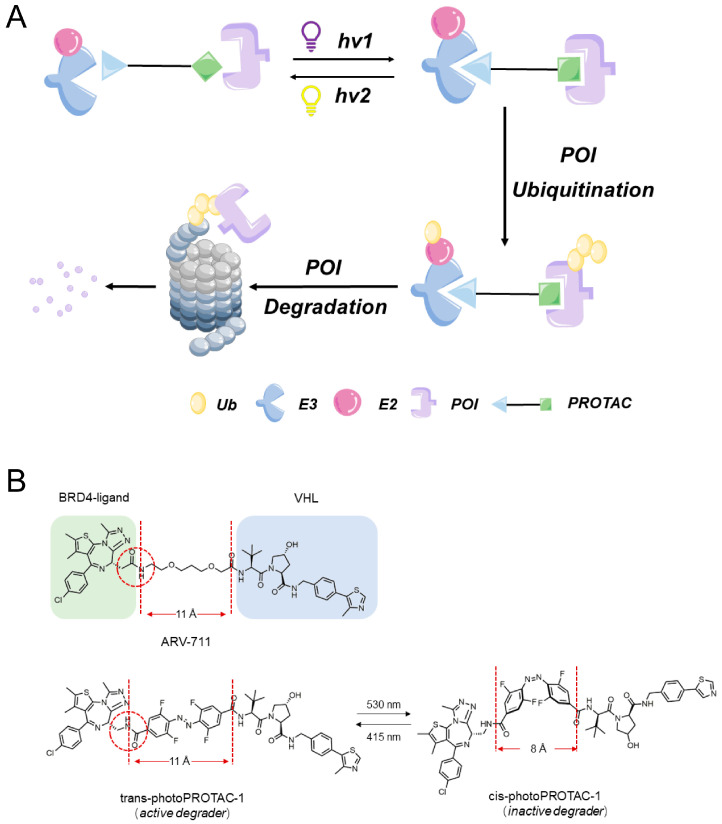
(**A**) Photoswitchable PROTACs. When exposed to two different specific wavelengths of light, photoPROTACs perform reversible photoisomerization between the “cis” and “trans” isomer forms, enabling precise and bidirectional control of the PROTACs. (**B**) Structure of highly active BET protein degrader ARV-771 displaying a maximal distance of 11 Å between both warhead moieties.

## Data Availability

Data available in a publicly accessible repository.

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
