# Peer review of "Recent Advances in Optically Controlled PROTAC"

_bioengineering, 2023, doi:10.3390/bioengineering10121368_

Round 1
Reviewer 1 Report
Comments and Suggestions for Authors
The manuscript titled "Recent Advances in optically controlled PROTAC" authored by Ouyang et al. offers a comprehensive overview of the latest developments in optically controlled PROTACs. PROTACs are widely acknowledged as a revolutionary innovation, offering solid approaches to protein degradation for therapeutic purposes. This review specifically delves into the recent advancements in optically controlled PROTACs, a topic of great interest within the scientific community. The manuscript is skillfully composed in the English language and thoroughly encompasses most of the key research findings related to optical PROTACs. In light of this, I strongly recommend publishing this manuscript with only minor revisions, as outlined in the following suggestions.
1. I kindly suggest including a recent study in chapter 3.2 of this review, as this particular work has received extensive citations and is highly illustrative of the themes related to photoswitchable PROTACs. https://pubs.acs.org/doi/10.1021/acs.jmedchem.9b02058
2. I kindly recommend presenting the chemical structures of the key optical PROTACs in chapters 3.1 and 3.2, if there are no copyright issues, as these chemical structures can be valuable for chemists when perusing the review.
3. Please thoroughly review the manuscript for possible typos and formatting issues. For instance, in line 214, "photoclavated" should be corrected to "photocleavable" or "photocleaved"; in line 238, "25μm" should be revised to "25 μm"; and in line 277, "1O2" should be corrected to "¹O₂".
Author Response
Dear Reviewer,
Thank you for your email dated Nov 11, 2023. We would like to express our sincere gratitude to the editor and all the reviewers for their constructive and encouraging comments. We have made significant improvements to our work and have included updated information to address the revisions suggested for our manuscript.
Attached to this email, you will find our detailed response to the comments raised by each reviewer. Additionally, we have provided a revised version of the manuscript, where all the changes have been clearly marked in red font.
- I kindly suggest including a recent study in chapter 3.2 of this review, as this particular work has received extensive citations and is highly illustrative of the themes related to photoswitchable PROTACs. https://pubs.acs.org/doi/10.1021/acs.jmedchem.9b02058
Response: Thanks a lot. We have added it to the text.
- I kindly recommend presenting the chemical structures of the key optical PROTACs in chapters 3.1 and 3.2, if there are no copyright issues, as these chemical structures can be valuable for chemists when perusing the review.
Response: Thanks a lot. We have added it to the text.
- Please thoroughly review the manuscript for possible typos and formatting issues. For instance, in line 214, "photoclavated" should be corrected to "photocleavable" or "photocleaved"; in line 238, "25μm" should be revised to "25 μm"; and in line 277, "1O2" should be corrected to "¹O₂".
Response: Thanks a lot. We have revised the related descriptions.
We would highly appreciate it if you could give a serious and timely consideration of this submission. Your assistance would be greatly valued. Look forward to hearing from you.
Yours sincerely
Ying Tan
21 Nov. 2023
Reviewer 2 Report
Comments and Suggestions for Authors
The authors provide a review on optically controlled PROTACs.
This is an interesting subunits or substantial interest and the authors do a reasonable job presenting the progress as well as discussing some of the limitations of the process. They include some more recent work, that provides an update compared to a more comprehensive and details review on the subject that was published in 2021 (PMID: 34115971).
The review would benefit from some textual work up to clarify or expand on specific items, some details provided in specific comments below. Also, the paper has four simple conceptual figures that are reasonable, although contain some errors. However, a paper on this subject should contain at least some molecular drawing of some of the compounds and how different groups change by light. It might not have to be as extensive as in the prior publication, but currently it comes across as very superficial in this regard.
Line 22 present due to low concentrations of targets lacking…?
Line 24 “high clinical effective doses”? high doses required to be clinically effective?
Line 24 expression? Or activity? function?
Line 25 drug resistance => Aren’t side effects relevant here as well?
Line 38 “at least four times to achieve” not necessarily, more accurate to replace this with: results in polyubiquitination?
Line 47 “highly developed” promising?
Line 49 delete “that complements the UPS pathway function”
Line 53 POI not defined yet I believe
Paragraph 48-56 seems to me that autophagy needs to be mentioned/defined briefly here.
Figure 1: Why start with polyubi in left top not free mono ubiquitin? ATP to AMP? Not ADP?
Line72 Delete”due to the fact that”
Line 72 E3 ligases only cytosolic? Really what reference? Maybe currently all PROTAC target E3s are cytosolic?
Line 74 what is allosteric interaction here? Does a first ligans site on the PROTAC trigger something for a second binding site?
Line 85-87 please explain event-driven vs occupancy driver.
Line 88 …than inhibiting individual…..?
Line 91 not sure what the 70 PROTAC targets precisely entails, where does this number come from? what is this group precisely?
Lin 97”induce a hook effect” please explain…
Line 127 ligand…ligand => correct?
At several places it states PROTAC ubiquitinates or PROTAC degrades… This is not correct…PROTAC induced degradation would be more accurate (e.g. line 141, 164, 177, 310)
Line 155 -156 difference between photocaged and switch needs to be explained here.
Line 167-171 Weird section of text to start 3.1.1. with? Wrong place?
Figure 3 add some example of molecular structures of PROTACs and how they change with light.
line 233 of small molecules of target proteins ?? confused reads that proteasome degrades small molecule? Does not make sense, please correct sentence.
Line 246 activatable cancer?
Line 253 typo?
Figure 4 no induction of what changes on PROTAC with hv1 or hv2… Also molecular example would be valuable
Line 323 deficits?
Line 327 and 328 both “Therefore,….”
Line 336 “problem” => Need?
Comments on the Quality of English Languagesee above
Author Response
Dear Reviewer,
Thank you for your email dated Nov 11, 2023. We would like to express our sincere gratitude to the editor and all the reviewers for their constructive and encouraging comments. We have made significant improvements to our work and have included updated information to address the revisions suggested for our manuscript.
Attached to this email, you will find our detailed response to the comments raised by each reviewer. Additionally, we have provided a revised version of the manuscript, where all the changes have been clearly marked in red font.
Thanks for your comments and approval, which are greatly appreciated.
The response to your comments is list below:
- Line 22 present due to low concentrations of targets lacking…?
Response: Thanks a lot. We have changed the related description.
- Line 24 “high clinical effective doses”? high doses required to be clinically effective?
Response: Thanks a lot. We have changed the related description.
- Line 24 expression? Or activity? function?
Response: Thanks a lot. We have changed the related description.
- Line 25 drug resistance => Aren’t side effects relevant here as well?
Response: Thanks a lot. We have changed the related description.
- Line 38 “at least four times to achieve” not necessarily, more accurate to replace this with: results in polyubiquitination?
Response: Thanks a lot. We have changed the related descriptions.
- Line 47 “highly developed” promising?
Response: Thanks a lot. We have changed the related descriptions.
- Line 49 delete“that complements the UPS pathway function”
Response: Thanks a lot. We have deleted.
- Line 53 POI not defined yet I believe
Response: Thanks a lot. The relevant definitions are given in line 24.
- Paragraph 48-56 seems to me that autophagy needs to be mentioned/defined briefly here.
Response: Thanks a lot. We have changed the related descriptions.
- Figure 1: Why start with polyubi in left top not free mono ubiquitin? ATP to AMP? Not ADP?
Response: Thanks a lot. We have changed the related figure. ATP-AMP: The literature related to UPS is ATP-AMP, which releases PPi. https://www.ncbi.nlm.nih.gov/pmc/articles/PMC7798376/
- Line72 Delete”due to the fact that”
Response: Thanks a lot. We have changed the related descriptions.
- Line 72 E3 ligases only cytosolic? Really what reference? Maybe currently all PROTAC target E3s are cytosolic?
Response: Thanks a lot. We have modified the related descriptions.
- Line 74 what is allosteric interaction here? Does a first ligans site on the PROTAC trigger something for a second binding site?
Response: Thanks a lot. The ternary, POI-PROTAC-E3, is very important for degradation. There is balance between these recognization.
- Line 85-87 please explain event-driven vs occupancy driver.
Response: Thanks a lot. We have changed the related descriptions. Most small molecule drugs need to bind to an enzyme or receptor at the active site in order to work, a mechanism of action known as the occuption-driven mode, but PROTAC can degrade a protein by grabbing it through any nook and cranny, it doesn't need to occupy an active pocket to work, and it doesn't need to rely on target occupancy to disrupt the function of the target protein. We call it an event-driven pharmacological mechanism.
- Line 88 …than inhibiting individual…..?
Response: Thanks a lot. It means that small molecule inhibitors target a specific domain of a protein.
- Line 91 not sure what the 70 PROTAC targets precisely entails, where does this number come from? what is this group precisely?
Response: Thanks a lot. We have modified the related descriptions.
- Lin 97”induce a hook effect” please explain…
Response: Thanks a lot. We have added it to the text. The phenomenon of false negative due to inappropriate ratio of antigen to antibody.
- Line 127 ligand…ligand => correct?
Response: Thanks a lot. We have changed the related descriptions.
- At several places it states PROTAC ubiquitinates or PROTAC degrades… This is not correct…PROTAC induced degradation would be more accurate (e.g. line 141, 164, 177, 310)
Response: Thanks a lot. We have changed the related descriptions.
- Line 155 -156 difference between photocaged and switch needs to be explained here.
Response: Thanks a lot. We have mentioned it in line 289-295.
- Line 167-171 Weird section of text to start 3.1.1. with? Wrong place?
Response: Thanks a lot. An introduction to the photocaged PROTACs was added in the front.
- Figure 3 add some example of molecular structures of PROTACs and how they change with light.
Response: Thanks a lot. We have added it to the text.
- line 233 of small molecules of target proteins ?? confused reads that proteasome degrades small molecule? Does not make sense, please correct sentence.
Response: Thanks a lot. We have changed the related descriptions.
- Line 246 activatable cancer?
Response: Thanks a lot. We have changed the related descriptions.
- Line 253 typo?
Response: Thanks a lot. We have changed the related descriptions.
- Figure 4 no induction of what changes on PROTAC with hv1 or hv2… Also molecular example would be valuable
Response: Thanks a lot. We have added it to the text.
- Line 323 deficits?
Response: Thanks a lot. Font changes have been made.
- Line 327 and 328 both “Therefore,….”
Response: Thanks a lot. We have changed the related descriptions.
- Line 336 “problem” => Need?
Response: Thanks a lot. We have changed the related descriptions.
We would highly appreciate it if you could give a serious and timely consideration of this submission. Your assistance would be greatly valued. Look forward to hearing from you.
Yours sincerely
Ying Tan
21 Nov. 2023